# A Case Study of Farmers' Behavioral Motivation Mechanisms to Crack the Fractal Multidimensional Relative Poverty Trap in Shaanxi, China

**Yao Zhang and Jianjun Huai \***

College of Economics and Management, Northwest A&F University, Xianyang 712100, China;
mzy2727@nwafu.edu.cn
**\*** Correspondence: hjj@nwsuaf.edu.cn

**Abstract:** China's approach to addressing rural poverty has evolved from a thorough resolution of absolute poverty to a focus on providing essential support for vulnerable individuals and improving the income and welfare conditions of those who are relatively poor, taking into account multiple dimensions. This study utilizes a dataset consisting of 526 research sets collected from the central region of Shaanxi Province. The research employs structural equation modeling to examine the fractal multidimensional relative poverty trap experienced by farm households. Additionally, the study investigates the behavior motivation mechanism that can potentially alleviate the multidimensional relative poverty trap at the farm household level. The study found that (1) farm households in the central Shaanxi region are caught in a multidimensional relative poverty trap, with education poverty and health poverty having a conduction and amplification effect; health poverty and education poverty amplify employment poverty; and consumption poverty amplifies education poverty and health poverty, and education poverty further amplifies information poverty. (2) Multidimensional relative poverty in farming households creates a self-reinforcing poverty trap, and community relative poverty amplifies the multidimensional poverty trap in farming households. (3) Farmers can overcome the multidimensional relative poverty trap through the behavior motivation mechanism.

**Keywords:** multidimensional relative poverty; poverty trap; behavior motivation





## 1. Introduction

The degree of poverty varies over time, depending on people's subjective and objective requirements as well as on their environment and social expectations. The United Nations has identified poverty eradication as the primary sustainable development goal, with the target year for achieving this objective set at 2030. In recent years, the Chinese government has implemented several measures to address the issue of poverty, resulting in a notable increase in the average income level among rural inhabitants. Consequently, significant progress has been made in alleviating absolute poverty, leading to a shift in focus toward addressing the issue of "relative poverty". However, the current focus lies on addressing the issue of relative poverty. Despite notable advancements, there are numerous obstacles to attaining inclusive social development, reducing the disparity in development between urban and rural areas and regions, and guaranteeing complete growth and equitable prosperity for all individuals [1]. China's poverty alleviation strategy has also shifted from realizing "Two Assurances and Three Guarantees" (no worries about food, no worries about clothing, and guarantees for compulsory education, basic medical care, and housing security) to alleviating the multidimensional relative poverty of unbalanced and inadequate development. The issue of multidimensional poverty within the population necessitates future attention, encompassing not only income poverty but also other elements of economic poverty and welfare poverty [2].

From a historical, geographic, environmental, and institutional perspective, it is evident that the income growth of Chinese poor people has been sluggish. Consequently, there has been a widening disparity in the standard of living and welfare between urban and rural areas. This unfortunate circumstance has resulted in a significant proportion of poor people experiencing relative poverty across various dimensions, including income, health, culture, and consumption [3,4]. In contrast, within the Loess Plateau region (located in the north-west of China), the presence of disparate socio-economic development, significant population outflow, and an aging demographic, together with the ongoing SARS-CoV pandemic over the past three years, have all contributed to a persistent and elevated state of multidimensional relative poverty among farming households residing in the area [5].

Academics summarize this state of poverty as a persistent and self-reinforcing trap. A number of studies have argued that external policy interventions have a long-term positive impact on breaking the poverty trap [6]. In addition, some scholars have emphasized that individual attitudes and perceptions are critical to cracking poverty [7]. It has also been found that there is a dynamic trap of socio-ecological interactions in the countryside [8]. Poverty traps manifest at both the individual and community levels among smallholder farmers residing in rural regions [9–12].

Studying the pathways and cross-scale effects within the multidimensional relative poverty trap requires consideration of three drivers [8]. Firstly, one of the reasons that can contribute to agricultural challenges is the individual farmer's lack of capability [13] and lack of initiative [14]. Second is the influence of environmental conditions, such as insufficient rural economic security, inadequate rural community cohesion, and inadequate rural industry integration [15]. Thirdly, due to policy factors, the necessity to continually adjust poverty governance policies aimed at enhancing farmers' prosperity in response to evolving conditions and temporal dynamics is evident [16]. For example, China's "Precision Poverty Alleviation" policy emphasizes precise identification, precise management, and precise assistance [17]. External policy incentives place more emphasis on income poverty alleviation, industrial poverty alleviation, and labor poverty alleviation [18], whereas top-down policy incentives lack the subjective and objective impact on smallholder farmers and other dimensions of poverty beyond income, making it difficult to create resilience among poverty farmers. The persistence of long-term, multidimensional relative poverty has resulted in the development of a poverty mentality among farmers. This mentality is characterized by the belief that the assets and resources they possess are insufficient to fulfill their actual needs or generate independent income. Consequently, farmers have come to perceive themselves as trapped in the predicament of multidimensional relative poverty, with little hope of escaping it. This poverty mentality can cause low-income people to fall into the relative poverty trap, i.e., to hesitate and wander and give up on getting out of poverty [19] and a lack of psychological capital such as the courage to get down to work and the confidence, optimism, and resilience to escape the trap of multidimensional relative poverty.

Some of the low-income farming families in Shaanxi have "resilience in adversity" and have the psychological and physical capital that helps them build their resistance to the complex threats they confront on a daily basis. The majority of multidimensional collectives comprised of farmers with limited financial resources often exhibit a lack of understanding regarding income generation and a lack of determination to overcome poverty. These groups tend to display cautiousness in their consumption and investment decisions, prioritizing immediate sustenance over long-term goals. Additionally, they have a tendency toward short-sightedness and a lack of ambition. Stimulated by the subsidy policy, some farmers are reluctant to leave their poverty status, and egalitarianism [20] may lead these farmers to take advantage of the loopholes in the policy and realize their own interests in the treatment of poor households; behind this phenomenon of voluntary poverty and unwillingness to lift themselves out of poverty is a lack of psychological resilience on the part of some rural households to lift themselves out of poverty. They lack the ability to

identify poverty, match resources, and choose actions to eliminate poverty, and they are "psychological poverty" because of their dependence on the culture of poverty [21].

Uncertainties, such as the instability of livelihoods and the variability of environmental conditions, might undermine the confidence of individual farmers and provide them with a quandary regarding their willingness to engage in proactive measures. Higher psychological resilience can help individuals cope with changing situations and strengthen their beliefs in the face of difficulties and risks. Enhancing psychological resilience requires the stimulation of personal effectiveness and the use of a strong will and a positive mindset to maximize one's own interests by integrating available resources, which is the first prerequisite for escaping from the multidimensional relative poverty trap. Enhance prospective productivity and well-being through the implementation of incentives that encourage individuals to reintegrate currently available resources. Developing strong endogenous dynamics for escaping multidimensional relative poverty requires improving the viability of farmers. This psychological expectation based on trust can improve the self-regulatory capacity of multidimensionally relatively poor farmers [22], and it can further prevent the reverse evolution of ability poverty into "spiritual poverty" [23]. On the other hand, an individual's efforts to escape poverty are also influenced by the community environment. Social welfare [24], public services [25], and financial services [26] in rural China are still poorly constructed, and as the income gap between urban and rural areas is widening, the multidimensional relative poverty in rural areas is unable to benefit from the trickle-down effect of economic growth. Farm households residing in underdeveloped rural communities face challenges in terms of access, security, and overall well-being [27]. These challenges are exacerbated by the absence of adequate community support, substandard infrastructure conditions, and the inadequate provision of public services. Consequently, farmers in such communities are constrained in their options for pursuing viable livelihood strategies.

To view multidimensional relative poverty among farm households simply as a problem addressed by one dimension and one mechanism, and to analyze only the interactions between the behavioral choices of farm households and their community environments, ignores the interplay between the poverty trap of the farm household and the poverty trap of the community. Improving the psychological resilience of farm households and establishing a perfect dynamic mechanism and welfare improvement mechanism for multidimensionally relatively poor farm households is the key to breaking the multidimensional relative poverty trap. In order to break the trap of low-level multidimensional relative poverty, it is necessary to stimulate farmers' personal capabilities, improve their initiative, stimulate the internal motivation of farmers to become rich and live happily, establish a stable and sustainable community mechanism for poverty eradication, and break the traps of "poverty of ability", "welfare poverty" and "spiritual poverty". Therefore, the research questions in this paper are: (1) What dimensions of relative poverty exist in Shaanxi farming households? (2) How is the multidimensional relative poverty trap formed, and what is its interaction mechanism? (3) How does behavior motivation mechanism action stimulate the personal effectiveness of multidimensional relative poverty farmers? (4) How does the behavior motivation mechanism action break the multidimensional relative poverty trap? Therefore, the contribution of this paper lies in verifying the self-reinforcing effect of the fractal multidimensional relative poverty trap in which Shaanxi farmers live and proposing a mechanism for farmers' behavior motivation mechanism action to crack the multidimensional relative poverty trap.

## 2. Theory and Assumptions

### 2.1. Relative Poverty Dimension

Relative poverty emphasizes that the poverty of individuals is below average. Multidimensional relative poverty includes dimensions such as "spatial poverty", "intergenerational poverty", and "hidden poverty" [28–30], whose dynamic characteristics need to be taken into account. Existing studies have examined the dimensions of income, education, health, housing, livelihood, and assets [3,31]; some scholars have also introduced

information technology levels [32] with infrastructure development [33], food nutrition structure [34], energy [35], and residential deprivation [36] dimensions. There is a downward trend in the multidimensional poverty level in rural areas of China, with obvious spatial disequilibrium [37]. In central Shaanxi, the economic income of farming households is relatively inadequate compared to the requirements of education, health, and overall well-being. This situation is influenced by historical, geographic, cultural, and economic factors; the development of farming households' communities is hindered by challenging natural conditions, remote geographic location, and regional development disparities (Table 1).

**Table 1.** Multidimensional relative poverty.

| First Dimension | Second Dimension | Connotation |
|---|---|---|
| Relative economic poverty | Relative income poverty | The income of farming households is less than 50% of the average income of society, reflecting the relative disparity in economic status and income inequality [38]. |
| Relative capability poverty | Relative education poverty | Insufficient and poor quality of public education resources in the rural areas where individuals or families live, even to the extent that they are unable to satisfy the need for all school-age members of local rural families to participate in and complete the compulsory stage of education, and there is a gap between them and the average level of education [39]. |
| | Relative employment poverty | Insufficient labor force, insufficient hours of work, high labor burden coefficient, inequality, and relative disparity in comparison with society as a whole or with specific groups in farm households [40]. |
| | Relative health poverty | The existence of sub-health or disease states among members of farming households, including physical and mental health poverty, reflects inequalities and relative disparities in the area of health [41]. |
| | Relative information poverty | Farmers still have a single way of obtaining outside information, a weak awareness of the use of new information, a low sensitivity to new information, and an insufficient ability to judge the truth or falsity of the information, reflecting the inequality and relative disparity in information acquisition and utilization [8]. |
| Relative welfare poverty | Relative consumption poverty | Family consumption of basic necessities, such as clothing, food, housing, and transportation, is lower than the average level of social consumption, making it difficult to meet the normal living needs of family members [42]. |

### 2.2. The Fractal Poverty Trap

The poverty trap is a state of persistent poverty that is autonomously reinforced at the individual or community level [8]. The poverty trap keeps the farm household or community in a stable and inefficient systemic equilibrium [43]. Therefore, escaping from relative poverty requires breaking this self-reinforcing equilibrium system and placing farmers or community groups in an alternative equilibrium with a higher sense of well-being [44,45].

Fractal Poverty Trap Theory explains the simultaneous occurrence of multiple levels of poverty traps [46]. Fractal poverty traps are poverty traps that are self-reinforcing with multiple dimensions, self-recycling and amplifying at different levels at the same time, and self-reinforcing through cross-feedback. Therefore, in lower productivity environments, dynamic poverty reduction focusing on only one dimension has little effect, while integrated interventions in all dimensions can reduce poverty significantly. Some farming households in central Shaanxi are in a fractal multidimensional relative poverty

trap. Farming households are confronted with various factors that contribute to their entrapment in multidimensional relative poverty. These factors include population aging, a limited labor force comprised of immediate family members, a high burden coefficient, possession of small plots of arable land that are of low quality, the poverty health status of household members, and a decline in income [47]. In addition, community factors such as industrial development, social network relationships [48], and infrastructural environments determine the relative poverty traps in the communities where Shaanxi farmers live. The multidimensional poverty trap arises from a combination of factors. Firstly, it is influenced by the value choices made by farmers, the accumulation of negative emotions within individuals, and the depletion of resources that are essential for farmers' livelihoods. Secondly, it is exacerbated by the inadequate development of infrastructure and public services, the degradation of the natural environment, and the unfavorable subsistence conditions prevalent in the communities where farmers reside. Ignoring the key cross-scale interactions between the relative poverty trap of the farm household and the relative poverty trap of the community may lead to a wrong assessment of multidimensional relative poverty, which in turn biases poverty reduction strategies and policy formulation. Therefore, in analyzing the fractal relative poverty trap in Shaanxi, the interplay of multidimensional relative poverty across levels should be emphasized. The self-reinforcing of the poverty trap posits that there exists a reciprocal relationship between multidimensional relative poverty in agricultural households and multidimensional relative poverty in communities, whereby each reinforces the other. Therefore, a hypothesis is postulated as follows:

**Hypothesis 1 (H1):** *The interaction between community multidimensional relative poverty and farm household multidimensional relative poverty results in a mutually reinforcing relationship.*

Self-recycling effects of poverty exist between different dimensions of the same dimension. If one dimension of multidimensional relative poverty in the same dimension is negatively correlated with another dimension of relative poverty, there is an offsetting effect of these dimensions of poverty; if different dimensions of relative poverty in the same dimension are positively correlated, there is a magnifying effect between them. For instance, farmers with limited financial resources possess a modest amount of livelihood capital. The majority of their available cash or savings is allocated towards meeting essential life necessities, leading to a state of relative poverty in various aspects such as education, skills, employment, and health. Conversely, farmers who possess stable and substantial cash reserves are more inclined to increase their consumption expenditures. This, in turn, elevates the overall consumption level of farmers, enhances the composition of their consumption patterns, and facilitates the transition from inflexible consumption to more adaptable forms of consumption [49]. Farmers engage in the diversification of their consumption patterns, encompassing sectors such as education and health, which consequently leads to a reduction in relative poverty levels, particularly in domains associated with subsistence consumption. Relatively affluent incomes can significantly improve the cognitive ability of farm households, thereby improving their relative educational poverty [50]. The higher the level of education of a farm household, the more livelihood options it has and the more income it generates from livelihood diversification; therefore, a reduction in relative poverty in terms of education is better able to reduce relative poverty in terms of income [51]. At the same time, education protects and improves farmers' own rights and interests, enhances life skills and their innovative capacity, promotes diversified employment of the labor force, and increases non-farm income [52]. Thus, reducing relative poverty in education means reducing relative poverty in employment. More income for farm households enhances their stock of assets such as physical capital, durables, and property [53], which in turn leads to better living conditions [54]. In conclusion, at the farm household level, there is a positive correlation between multidimensional relative poverty. Therefore, a hypothesis is postulated as follows:

**Hypothesis 2 (H2):** *There is an amplification effect within the multidimensional relative poverty of farm households.*

*2.3. Mechanisms of Farmers' Behavior Motivation to Crack the Multidimensional Poverty Trap*

The behavior motivation mechanism consists of two components: personal effectiveness and future value orientation. Personal effectiveness is one of the three elements of an individual's mechanism to address poverty [55] and refers to the ability of a farmer to change his or her poverty situation through his or her own actions and efforts. This mechanism places significant emphasis on the active engagement of economically disadvantaged farmers with diverse dimensions in enhancing their socio-economic status. It also aims to regulate the efficacy of farmers in stimulating their own productivity, thereby establishing a foundation for their involvement in poverty alleviation initiatives and preventing relapse into poverty. Due to a deficiency in inherent motivation within the rural impoverished population, rural households have experienced a reduction in their needs, a decline in cultural practices, diminished confidence, inadequate commitment to combat poverty and pursue independent development, and a persistent reliance on external assistance through the mindset of "waiting, relying, and asking for help". Consequently, their behavioral initiative is insufficient, and their sense of participation is weak. Behavior motivation mechanisms reduce the risks and losses associated with shocks by mobilizing farmers' own personal effectiveness to adapt to new modes of production activities, policies, and environments and to change adaptive behaviors in the face of shocks [56] and losses [57]. Therefore, the long-term income-generating mechanism for low-income farmers is a mechanism for farmers to adapt on their own under the government's program adaptation and to realize farmers' income increase by improving their livelihood capital [58]. The effective resolution of the multidimensional relative poverty trap is contingent upon a substantial reliance on ambition and proactive measures to elevate individuals from impoverished conditions [59]. Enhancing "income aspirations" constitutes a significant element of individual initiative, as it can serve as a driving force behind investment behavior, engagement in non-agricultural jobs, and consequent alleviation of poverty within farm households and families.

Existing research on the subjective factors influencing poverty highlights the significance of the future value orientation of impoverished populations. This research underscores the notion that individuals in poverty, such as farmers, engage in decision-making processes that enable them to break free from multidimensional relative poverty. These decision-making processes involve carefully considering various options and opportunities, as well as making predictions and formulating plans based on anticipated future outcomes [60]. The future value orientation requires that multidimensional groups of relatively poor people have long-term goals and visions for their productive lives, are able to grasp the state of their future economies, and are able to make clear and rational decisions in pursuit of their future goals. If multidimensional groups of relatively poor people believe they have the ability to change their lives, have a clear goal, have a high level of psychological resilience, and have confidence in their future prospects as well as an unambiguous understanding of them, then they may be more motivated to pursue that goal and be willing to change the status quo. On the contrary, if they do not have clear goals for the future and are not confident that they will be able to achieve them, they may give up and are more likely to fall into the poverty trap, which also implies a lack of psychological resilience. Personal efficacy beliefs, personal control beliefs, fear of failure, and future development planning influence poverty people to increase production and income [61]. Therefore, the mechanism of behavior motivation in this paper can stimulate personal effectiveness, increase the willingness to get rid of poverty, enhance belief in action, reduce concern about risk, increase the future value orientation to plan for the future and help farmers choose familiar paths to get rid of poverty in advance. Therefore, a hypothesis is postulated as follows (Figure 1):

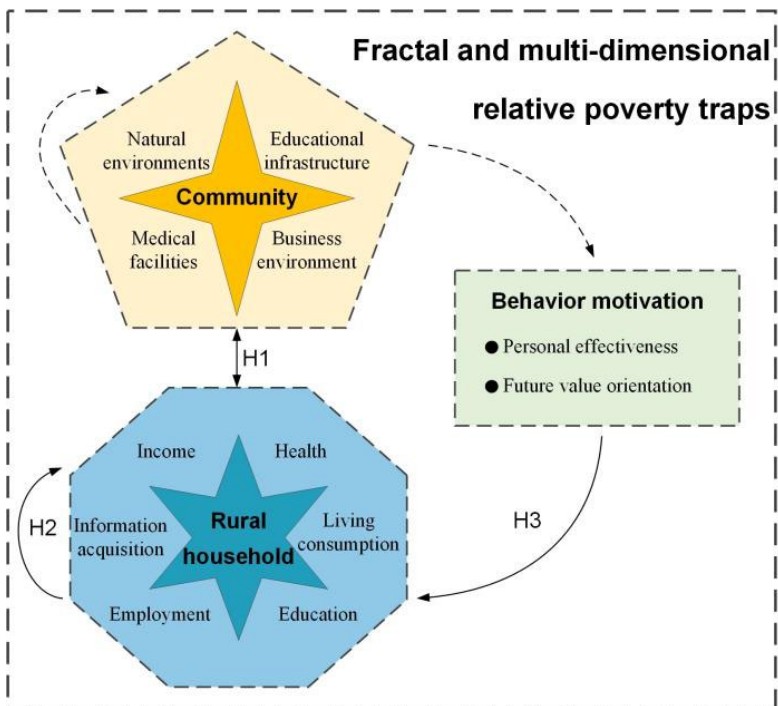

**Figure 1.** A theoretical framework for farmers' behavior motivation to crack the farmer-community fractal multidimensional relative poverty trap. Note: The arrows in the figure indicate poverty self-reinforcing effects, the solid arrows indicate the hypotheses to be tested, and the related concepts are shown in the list of variables. The yellow color represents the multidimensional relative poverty trap of the community, the blue part indicates the multidimensional relative poverty trap of the farm household, and the green part represents the behavior motivation mechanism.

**Hypothesis 3 (H3):** *Behavior Motivation mechanisms are effective in reducing multidimensional relative poverty among farm households.*

## 3. Methodology

### 3.1. Data

In this paper, based on the documents (The aforementioned documents include the Opinions on the Establishment of a Poverty Withdrawal Mechanism and the Implementation Measures for the Special Assessment and Inspection of the Withdrawal of Poverty Counties in Shaanxi Province.) presented by the Shaanxi Provincial Government, the last exited poverty counties in Shaanxi Province were selected as the study area, covering 10 districts and counties in the northern part of Guanzhong, Shaanxi, including Fufeng County, Qianyang County, Long County, Linyou County, Xunyi County, Baishui County, Pucheng County, Fengxiang District of Baoji City, Yaozhou District of Tongchuan City, and Yintai District (as shown in Figure 2 below). We used regional stratified sampling with the sample random sampling method; under each district and county, we randomly selected three towns, and under each town, we randomly selected three villages and conducted household face-to-face interview questionnaire research. Based on the proposed calculation of the income-relative poverty line based on a fixed proportional value of 50 percent of the median disposable income, income-related poverty is used as the threshold for determining multidimensional relative poverty for farm households. That is when income relative poverty exists, then individuals are in multidimensional relative poverty. We collected 600 questionnaires in May–August 2022 and screened 526 valid questionnaires from multidimensional relative poverty groups for empirical analysis in this paper.

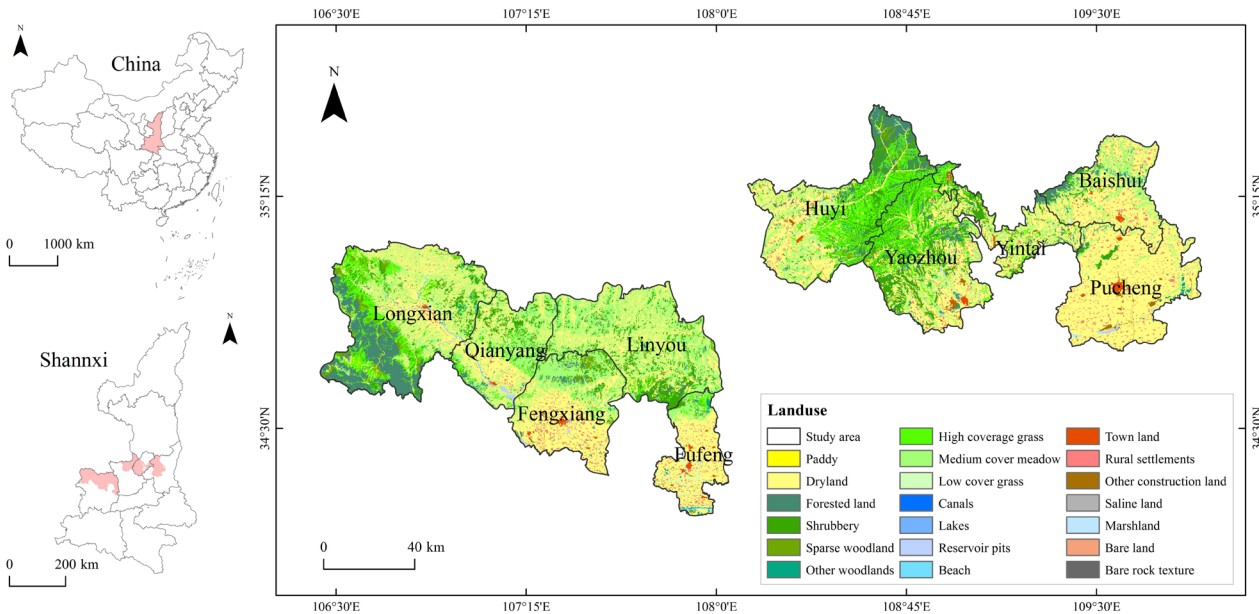

**Figure 2.** Study area located in central Shaanxi.

### 3.2. Measurement Scale

In order to assess the multidimensional relative poverty of farm households, we employ data pertaining to income, employment, health, education, access to information, and consumption. Similarly, to gauge the multidimensional relative poverty of the community, we consider factors such as the natural environment, local education and medical care, and the local business environment. The measurement indicators were assessed using Likert five-point scales. The descriptive statistical analysis of the entire dataset (Table 2) reveals that the average values for income, information, and living consumption among farm households are comparatively low. Conversely, the levels of employment, health, and the natural environment within the community, as well as the personal effectiveness and future value orientation of farm households, are relatively high.

**Table 2.** Measurement scale and descriptive statistical analysis.

| Latent Variable | Indicator Description | Five-Point Likert Scale | Mean | SD | Median |
|---|---|---|---|---|---|
| Income | Annual disposable household income (IN1) | 1 = ¥0–¥1400; 2 = ¥1400–¥2800; 3 = ¥2800–¥4200; 4 = ¥4200–¥5600; 5 = ¥5600–¥7000 (RMB) | 2.049 | 1.213 | 2.000 |
| | Annual household income from agriculture (IN2) | 1 = ¥0–¥1000; 2 = ¥2000–¥3000; 3 = ¥3000–¥4000; 4 = ¥4000–¥5000; 5 = ¥5000–¥6000 (RMB) | 2.103 | 1.170 | 2.000 |
| | Annual household income from employment (IN3) | 1 = ¥0–¥1000; 2 = ¥2000–¥3000; 3 = ¥3000–¥4000; 4 = ¥4000–¥5000; 5 = ¥5000–¥6000 (RMB) | 1.922 | 1.141 | 2.000 |
| | Discounted value of the family's main real estate, car, and financial assets (IN4) | 1 = 0–1; 2 = 1–2; 3 = 2–3; 4 = 3–4; 5 = over 4 (Ten Thousand RMB) | 2.279 | 1.151 | 2.000 |
| Employment | Number of family laborers (EM1) | 1 = 0; 2 = 1; 3 = 2; 4 = 3; 5 = over 3 | 3.951 | 0.988 | 4.000 |
| | Percentage of labor hours per year for household heads (EM2) | 1 = 0–20%; 2 = 20–40%; 3 = 40–60%; 4 = 60–80%; 5 = 80–100% | 4.139 | 0.895 | 4.000 |
| | Household labor burden ratio (EM3) | 1 = 0–20%; 2 = 20–40%; 3 = 40–60%; 4 = 60–80%; 5 = 80–100% | 3.850 | 1.028 | 4.000 |

**Table 2.** *Cont.*

| Latent Variable | Indicator Description | Five-Point Likert Scale | Mean | SD | Median |
|---|---|---|---|---|---|
| Information | Number of agricultural materials purchased through the Internet (INF1) | 1 = 0; 2 = 1; 3 = 2; 4 = 3; 5 = over 3 | 1.677 | 0.935 | 1.000 |
| | Amount of agricultural materials purchased by households through the Internet (INF2) | 1 = ¥0–¥50; 2 = ¥50–¥100; 3 = ¥100–¥150; 4 = ¥150–¥200; 5 = over ¥200 (RMB) | 1.819 | 0.972 | 2.000 |
| | Household access to agricultural information (INF3) | Number of devices accessing information 1 = 0; 2 = 1; 3 = 2; 4 = 3; 5 = over 3 | 2.150 | 1.059 | 2.000 |
| Education | Number of family members completing primary education (ED1) | 1 = 0; 2 = 1; 3 = 2; 4 = 3; 5 = over 3 | 2.935 | 1.222 | 3.000 |
| | Family education budget (ED2) | 1 = Spend very little; 2 = Lower spending; 3 = Moderate cost; 4 = More expenses; 5 = Spending a lot | 2.504 | 1.315 | 2.000 |
| | Number of family members completing junior high school (ED3) | 1 = 0; 2 = 1; 3 = 2; 4 = 3; 5 = over 3 | 2.641 | 1.263 | 2.000 |
| | Educational level of the head of household (ED4) | 1 = Illiterate; 2 = Primary school; 3 = Junior high school; 4 = High school; 5 = Bachelor's degree or above | 2.359 | 1.281 | 2.000 |
| Consumption of life | Monthly household expenditure on travel (CL1) | 1 = Spend very little; 2 = Lower spending; 3 = Moderate cost; 4 = More expenses; 5 = Spending a lot | 1.951 | 1.028 | 2.000 |
| | Monthly household expenditure on fuel (CL2) | As above | 2.338 | 1.211 | 2.000 |
| | Monthly household spending on clothing (CL3) | As above | 1.589 | 0.857 | 1.000 |
| | Monthly household expenditure on food (CL4) | As above | 2.106 | 1.058 | 2.000 |
| Health | Household spending on medical care in the past year (HE1) | 1 = Spending more than ¥5000; 2 = Spending ¥4000–¥5000; 3 = Spending ¥3000–¥4000; 4 = Spending ¥3000–¥2000; 5 = Spending less than ¥2000 | 2.648 | 1.197 | 2.000 |
| | Number of persons in the household who are incapable of self-care due to illness or disability (HE2) | 1 = 4 and above; 2 = 3; 3 = 2; 4 = 1; 5 = 0 | 3.846 | 0.847 | 4.000 |
| | Number of persons living in households with chronic diseases (HE3) | 1 = 4 and above; 2 = 3; 3 = 2; 4 = 1; 5 = 0 | 2.243 | 1.162 | 2.000 |
| Natural resources | Percentage of household crops affected (NR1) | 1 = 0–20%; 2 = 20–40%; 3 = 40–60%; 4 = 60–80%; 5 = 80–100% | 3.570 | 1.063 | 4.000 |
| | Cultivated land at the disposal of the household (NR2) | 1 = 0–0.82 acres; 2 = 0.82–1.65 acres; 3 = 1.65–2.47 acres; 4 = 2.47–3.29 acres; 5 = more than 3.29 acres | 3.312 | 1.079 | 3.000 |
| | Level of green cover around household cultivated land (NR3) | 1 = Strongly disagree; 2 = Quite disagree; 3 = Neutral; 4 = Quite agree; 5 = Strongly agree | 3.738 | 1.072 | 4.000 |
| | Duration of crop damage (NR4) | 1 = 0–5 days; 2 = 5–10days; 3 = 10–15 days; 4 = 15–20 days; 5 = over 20 days; | 2.323 | 1.105 | 2.000 |
| | Types of crops affected (NR5) | 1 = 0 disasters; 2 = 1 disaster; 3 = 2 disasters; 4 = 3 disasters; 5 = more than 3 disasters | 2.108 | 1.073 | 2.000 |

**Table 2.** *Cont.*

| Latent Variable | Indicator Description | Five-Point Likert Scale | Mean | SD | Median |
|---|---|---|---|---|---|
| Educational institutions | There are enough schools in your neighborhood to meet your needs (EI1) | 1 = Strongly disagree; 2 = Quite disagree; 3 = Neutral; 4 = Quite agree; 5 = Strongly agree | 1.968 | 1.041 | 2.000 |
| | The educational institution attended by the family member boasts a very competent faculty (EI2) | As above | 2.414 | 1.161 | 2.000 |
| | The school in which the family member resides has a commendable standard of e-learning. (EI3) | As above | 2.243 | 1.093 | 2.000 |
| | The provision of educational enrichment within the school attended by family members. (EI4) | As above | 2.198 | 1.061 | 2.000 |
| Medical institution | The quality of service provided by the physicians at the hospital that your family regularly visits is of exceptional standard. (MI1) | As above | 3.228 | 1.123 | 3.000 |
| | The quality of medical facilities provided by the hospital catering to your family is of a commendable standard. (MI2) | As above | 3.017 | 1.112 | 3.000 |
| | There are enough hospitals, clinics, and health centers near your home (MI3) | As above | 2.850 | 1.120 | 3.000 |
| Business environment | Number of people in your household working in the tertiary sector in the local area (BE1) | 1 = 0; 2 = 1; 3 = 2; 4 = 3; 5 = over 3 | 1.899 | 0.978 | 2.000 |
| | Number of people in your household working in the secondary sector in your local area (BE2) | As above | 2.502 | 1.109 | 2.000 |
| | Sufficient number of rural institutions, and cooperative organizations in the village (BE3) | 1 = Strongly disagree; 2 = Quite disagree; 3 = Neutral; 4 = Quite agree; 5 = Strongly agree | 2.703 | 1.104 | 3.000 |
| | The village has a large area of cooperative facility agriculture, shared farm machinery and equipment (BE4) | As above | 1.802 | 0.962 | 2.000 |
| Personal effectiveness | Regarding farming, you are in a positive state of mind. (PE1) | As above | 3.435 | 1.100 | 3.000 |
| | Throughout the agricultural production process, you have the ability to continuously inspire yourself. (PE2) | As above | 3.390 | 1.054 | 3.000 |
| | You've been able to maintain a sense of normalcy in your productive life (PE3) | As above | 3.819 | 1.067 | 4.000 |
| | You are able to be conscientious in the agricultural process (PE4) | As above | 3.517 | 1.041 | 4.000 |
| Future value orientation | You are certain to continue your current production model in the future and are at an advantage (FO1) | As above | 3.930 | 1.033 | 4.000 |
| | You are optimistic about the future of the agricultural production you are currently engaged in (FO2) | As above | 3.502 | 1.085 | 4.000 |
| | You have a clear plan for your future production strategy (FO3) | As above | 3.593 | 1.089 | 4.000 |
| | You have a solid understanding of the agricultural industry's future growth (FO4) | As above | 3.152 | 1.088 | 3.000 |

### 3.3. Model Setting

Based on the covariance matrix of variables, structural equation modeling can effectively analyze the structural relationship between latent variables that cannot be directly

observed, and it has gradually become one of the most important research methods in economics and management. The types of indicators in structural equation modeling usually include reflective and formative types, but if reflective and formative indicators are misused, it will lead to bias in parameter estimation. Reflective structural equations need to satisfy five criteria for modeling: (1) Causality needs to be from latent variables to observed variables. (2) The observed variables must be internally consistent. (3) Observed variables need to be moderately to highly correlated. (4) A latent variable must have at least three observed variables. (5) Removing a particular observed variable from a latent variable does not affect the significance of the latent variable. In this paper, changes in the poverty level of some dimensions will lead to changes in the corresponding poverty characteristics of individuals or communities; for example, a bad business environment will affect the development of local rural institutions, community rural cooperatives, and agricultural machinery sharing organizations, which is contrary to the principle of formative structural equations, i.e., changes in latent variables will not lead to changes in the observed variables [62]. Therefore, with reference to the above criteria, the relationship between observed and latent variables in this paper is more applicable to reflective measurement models.

This paper constructs a fractal multidimensional relative poverty trap structural equation model (Model I) and a farmer's behavior motivation cracking multidimensional relative poverty trap structural equation model (Model II), respectively. Model I aims to test hypotheses one (H1) and two (H2) by demonstrating the existence of amplification and cross-layer transmission effects within the fractal multidimensional relative poverty trap. Model II aims to test hypothesis three (H3), which proves that the behavior motivation mechanism is effective in cracking the multidimensional relative poverty trap of farm households. Structural equation modeling can better explain the relationship between different latent variables that are difficult to observe directly compared to ordinary regression models, as a way to better analyze the amplifying or offsetting effects between different dimensions of relative poverty.

### 3.4. K-Mean Clustering and Multicluster Analysis

In the application of reflective structural equation modeling, the importance of multicluster analysis cannot be ignored. Its role lies in (1) the validation of model robustness, assessing whether the model is consistent across different clusters. (2) Enhanced theoretical generalizability, which enhances the generalizability of a model if it is supported across multiple clusters. (3) Multi-cluster analysis can deepen research insights into important associations that may exist within specific subclusters. Multi-cluster analysis provides researchers with the opportunity to explore these relationships in depth within specific clusters. Therefore, multi-cluster analysis requires the data to be classified or clustered first. Clustering offers greater flexibility than classification, allowing researchers to explore and understand the patterns inherent in data without pre-conditions. This delineation can be done based on requirements or modeling needs, or it can simply help us explore the natural structure and distribution of the data without relying on a priori labels or classifications, thus reducing subjective bias [63]. K-mean clustering can group multiple sets of data based on their characteristics. This type of method needs to find a number of random clustering centers, then, according to the distance between each data point and the center of some of these clusters, decide which data points are suitable for the same group, and then, according to these groups, get the new center of clusters, reuse the new center of clusters to correct the results of clustering, repeat the implementation of these steps until the termination of the set conditions is met, and finally get the clustering results successfully. This algorithm belongs to the most typical segmented clustering algorithms, where each data point, the distance from the center of the cluster can have the smallest squared error. Assuming that the individual endowment of a group of farmers has $h$ cluster centers, where the $k$th cluster can be represented by the set $Gk$, $\mu$ is the center point in the

cluster, assuming that the clustering cluster G$k$ $\{x_{1k}, x_{2K}, x_{3K}, \ldots\ldots x_{nk}\}$ contains n$k$ sets of data, then the squared error $e_k$ of this cluster can be defined as follows:

$$e_k = \sum_{i} |x_{ik} - \mu_k|^2 \tag{1}$$

which then yields the summed squared error E of the number of *h* clusters:

$$E = \sum_{k=1}^{h} e_k^2 \tag{2}$$

The overall sum of the squares of the groups gets smaller and smaller as the center point keeps changing. When the iteration ends and the sum of squares of the cluster reaches the minimum value, the center point no longer changes, and the grouped cluster at this point is the one we need.

## 4. Results

### 4.1. Reliability and Validity Tests

To ensure the reliability and validity of the findings, we first tested the model for reliability and validity. We used the alpha reliability coefficient method to evaluate the reliability of the latent and observed variables required for the model, which, according to the existing criteria, indicates that the internal consistency of the scale is good and the reliability of the data is high [64]. In Table 3, the "Cronbach's alpha If Item Deleted" test shows that there is no significant increase in the reliability coefficient if any item is deleted, thus indicating that the item should not be deleted. The CITC values for each question item were greater than 0.6, indicating a good correlation between the analyzed items as well as a good level of reliability. KMO value and Bartlett's spherical test were carried out, and the KMO value was calculated to be 0.789, and the significance of Bartlett's spherical test was 0.000, which passed the 1% significance test, indicating that the scale can be analyzed by factor analysis, and therefore the original hypothesis that the perturbation terms in the equations are independent of one another can be rejected, which proves the applicability of structural equations. Validated factor analysis of the 12 selected factors showed that all 12 factors corresponded to AVE values greater than 0.6 and all CR values higher than 0.8, implying that the data from this analysis had good convergent validity. Therefore, the measurement model passed the validity test.

**Table 3.** Reliability and Validity tests.

| Variable | Indicator Description | Corrected Item-Total Correlation (CITC) | Cronbach's Alpha If Item Deleted | Cronbach $\alpha$ | Average Variance Extracted | Combinatorial Reliability |
|---|---|---|---|---|---|---|
| Income1 | IN1 | 0.820 | 0.877 | | | |
| Income2 | IN2 | 0.804 | 0.882 | | | |
| Income3 | IN3 | 0.815 | 0.879 | 0.911 | 0.721 | 0.912 |
| Income4 | IN4 | 0.752 | 0.900 | | | |
| Employment1 | EM1 | 0.735 | 0.761 | | | |
| Employment2 | EM2 | 0.677 | 0.819 | 0.845 | 0.649 | 0.847 |
| Employment3 | EM3 | 0.731 | 0.767 | | | |
| Information1 | INF1 | 0.680 | 0.737 | | | |
| Information2 | INF2 | 0.694 | 0.720 | 0.816 | 0.603 | 0.820 |
| Information3 | INF3 | 0.634 | 0.786 | | | |
| Education1 | ED1 | 0.733 | 0.888 | | | |
| Education2 | ED2 | 0.806 | 0.862 | | | |
| Education3 | ED3 | 0.805 | 0.862 | 0.901 | 0.696 | 0.902 |
| Education4 | ED4 | 0.770 | 0.875 | | | |

**Table 3.** *Cont.*

| Variable | Indicator Description | Corrected Item-Total Correlation (CITC) | Cronbach's Alpha If Item Deleted | Cronbach α | Average Variance Extracted | Combinatorial Reliability |
|---|---|---|---|---|---|---|
| Consumption of life1 | LI1 | 0.674 | 0.858 | | | |
| Consumption of life2 | LI2 | 0.790 | 0.816 | 0.873 | 0.649 | 0.880 |
| Consumption of life3 | LI3 | 0.764 | 0.834 | | | |
| Consumption of life4 | LI4 | 0.724 | 0.839 | | | |
| Health1 | HE1 | 0.730 | 0.721 | | | |
| Health2 | HE2 | 0.669 | 0.802 | 0.827 | 0.633 | 0.838 |
| Health3 | HE3 | 0.704 | 0.746 | | | |
| Natural resources1 | NR1 | 0.752 | 0.881 | | | |
| Natural resources2 | NR2 | 0.708 | 0.890 | | | |
| Natural resources3 | NR3 | 0.764 | 0.878 | 0.902 | 0.649 | 0.902 |
| Natural resources4 | NR4 | 0.798 | 0.870 | | | |
| Natural resources5 | NR5 | 0.752 | 0.881 | | | |
| Educational institutions1 | EI1 | 0.707 | 0.819 | | | |
| Educational institutions2 | EI2 | 0.706 | 0.821 | 0.859 | 0.605 | 0.860 |
| Educational institutions3 | EI3 | 0.716 | 0.815 | | | |
| Educational institutions4 | EI4 | 0.690 | 0.826 | | | |
| Medical institution1 | MI1 | 0.664 | 0.777 | | | |
| Medical institution2 | MI2 | 0.693 | 0.747 | 0.825 | 0.612 | 0.826 |
| Medical institution3 | MI3 | 0.687 | 0.753 | | | |
| Business environment1 | BE1 | 0.717 | 0.833 | | | |
| Business environment2 | BE2 | 0.699 | 0.841 | 0.868 | 0.626 | 0.870 |
| Business environment3 | BE3 | 0.733 | 0.826 | | | |
| Business environment4 | BE4 | 0.737 | 0.826 | | | |
| Personal effectiveness1 | PE1 | 0.745 | 0.830 | | | |
| Personal effectiveness2 | PE2 | 0.706 | 0.846 | 0.873 | 0.634 | 0.874 |
| Personal effectiveness3 | PE3 | 0.700 | 0.848 | | | |
| Personal effectiveness4 | PE4 | 0.761 | 0.824 | | | |
| Future value orientation1 | FO1 | 0.745 | 0.838 | | | |
| Future value orientation2 | FO2 | 0.723 | 0.846 | 0.876 | 0.641 | 0.877 |
| Future value orientation3 | FO3 | 0.762 | 0.830 | | | |
| Future value orientation4 | FO4 | 0.707 | 0.852 | | | |

Standardized Cronbach α: 0.837.

### 4.2. Model Fit Test

The overall model fitness test index of structural equation modeling mainly includes Absolute Goodness-of-Fit Indices and Value-Added Fitness Index. According to the test results in Table 4, $\chi^2/df < 3$ as well as CFI, TLI, and IFI were all greater than 0.9, so the overall model fit was good. In addition, the obtained Variance Inflation Factors of the observed samples are all less than 10, indicating that there is no multicollinearity between the measured variables.

**Table 4.** Model Fit Test.

| Model | $\chi^2$ | *df* | *p* | $\chi^2/df$ | GFI | RMSEA | CFI | TLI | IFI |
|---|---|---|---|---|---|---|---|---|---|
| Model I | 1589.605 | 879 | >0.050 | <31.808 | 0.872 | 0.039 | 0.951 | 0.945 | 0.951 |
| Model II | 1051 | 367 | >0.050 | <32.865 | 0.877 | 0.060 | 0.925 | 0.917 | 0.925 |

### 4.3. Structural Equation Results

4.3.1. The Interdimensional Role of Fractal Poverty

Structural equation model I (shown in Figures 3 and 4, Table 5) is a multidimensional relative poverty internal amplification network as well as a cross-layer interaction network. At the community level, the magnifying effect of the natural level on income is 0.314, the magnifying effect of educational facilities and education is 0.309, the narrowing effect of medical facilities and health is 0.303, and the magnifying effect of doing business level on

employment is 0.151. This suggests that rural community multidimensional traps in central Shaanxi amplify farmer multidimensional traps, which confirms hypothesis H1. From the model, there is a significant mediating effect between the dimensions of the farm household, especially the three dimensions of income, health, and labor, which are key nodes in the self-reinforcing of the multidimensional relative poverty trap of the farm household. The difference is that there is no significant amplification effect among the community-relative poverty dimensions, which may be because the community-relative poverty dimensions are relatively independent of each other, and there is no strong collaboration and bonding among the natural resources, educational institutions, medical institution and business environments of the community.

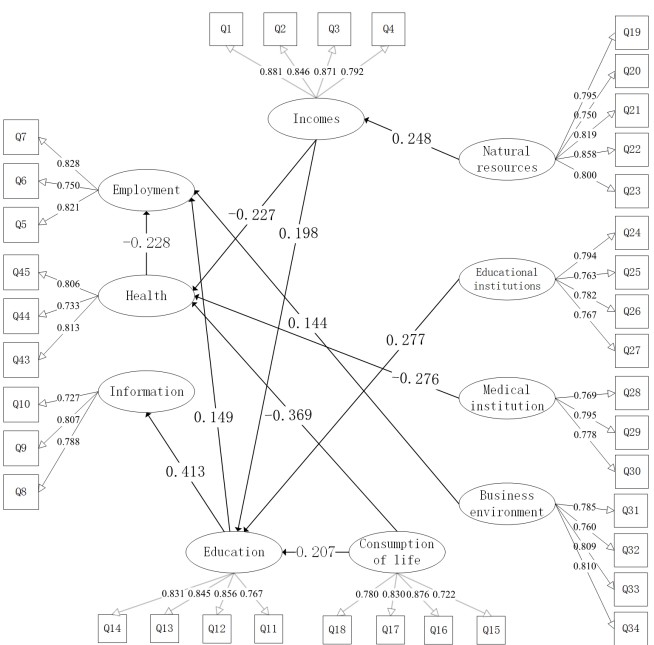

**Figure 3.** Structural equation modeling of the individual-community fractal multidimensional relative poverty trap (Model I).

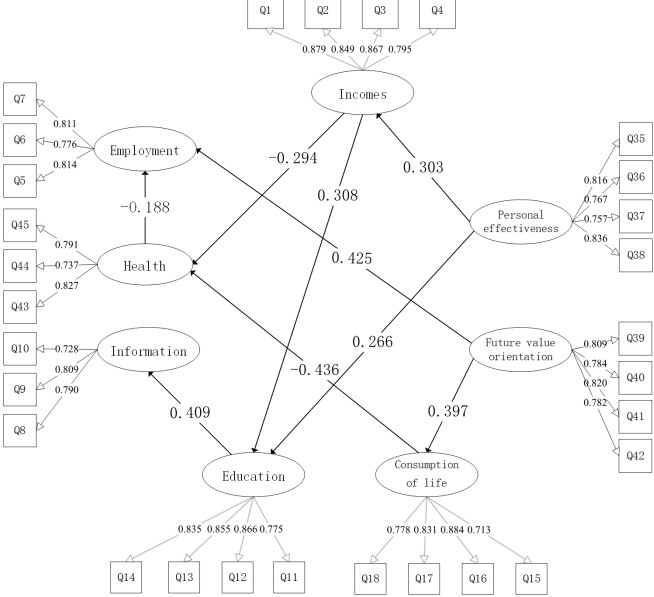

**Figure 4.** Structural Equation modeling of behavior motivation on multidimensional relative poverty cracking mechanisms (Model II).

**Table 5.** Path coefficients and significance among latent variables.

| | Pathway | Estimate | S.E. | C.R. | *p* | Std. Est |
|---|---|---|---|---|---|---|
| | IN←NR | 0.248 | 0.060 | 5.209 | *** | 0.248 |
| | HE←IN | −0.227 | 0.039 | −5.192 | *** | −0.227 |
| | ED←IN | 0.198 | 0.038 | 4.513 | *** | 0.198 |
| | ED←CL | 0.207 | 0.075 | 3.429 | *** | 0.207 |
| | HE←CL | −0.369 | 0.065 | −7.248 | *** | −0.369 |
| Model I | HE←MI | −0.276 | 0.056 | −5.415 | *** | −0.276 |
| | ED←EI | 0.277 | 0.069 | 4.487 | *** | 0.277 |
| | EM←HE | −0.270 | 0.046 | −4.989 | *** | −0.270 |
| | INF←ED | 0.413 | 0.041 | 7.964 | *** | 0.413 |
| | EM←ED | 0.149 | 0.044 | 2.969 | 0.003 | 0.149 |
| | EM←BE | 0.144 | 0.055 | 2.759 | 0.006 | 0.144 |
| | CL←FO | 0.397 | 0.044 | 7.851 | *** | 0.397 |
| | IN←PE | 0.303 | 0.057 | 6.293 | *** | 0.303 |
| | ED←IN | 0.308 | 0.042 | 6.444 | *** | 0.308 |
| | HE←IN | −0.294 | 0.041 | −6.469 | *** | −0.294 |
| Model II | HE←CL | −0.436 | 0.066 | −8.688 | *** | −0.436 |
| | ED←PE | 0.266 | 0.051 | 5.488 | *** | 0.266 |
| | EM←HE | −0.188 | 0.04 | −3.839 | *** | −0.188 |
| | INF←ED | 0.409 | 0.04 | 7.938 | *** | 0.409 |
| | EM←FO | 0.425 | 0.048 | 8.390 | *** | 0.425 |

Note: *** is significant at the 1% levels.

At the farm household, IN amplifies HE and thus EM, such that a 1 standard deviation increase in IN will increase HE by 0.227 standard deviations (as shown in Table 5), and a 1 standard deviation increase in HE will increase EM by 0.270 standard deviations. Therefore, in the single chain "IN→HE→EM", the indirect amplification effect of farm household income on employment is 0.061 (0.227 ∗ 0.270 = 0.061). Second, in the single chain of "IN→ED→EM", the indirect amplification effect of the multidimensionality of farm households is 0.030 (0.198 ∗ 0.149 = 0.030). Finally, in the single chain of "CL→ED →INF", the indirect amplification effect of the multidimensionality of farmers is 0.085 (0.207 ∗ 0.413 = 0.085); the health amplifies the employment effect is 0.270. This suggests that there is a multidimensional amplification trap at the farm household level for farmers in central Shaanxi, which confirms hypothesis H2.

4.3.2. Behavior Motivation Mechanism

From Model II, personal effectiveness has an amplification effect of 0.303 on income and 0.266 on education, with further transmission to health and employment. This implies that when the subjective will of the population to alleviate multidimensional relative poverty strengthens, their sense of responsibility and awareness increases, intensifying the imperative to enhance their income and well-being and bolstering their psychological resilience. Individual Behavior Motivation can stimulate the sense of employment and income of multidimensionally relatively poor farmers to alter their state of multidimensional relative poverty by actively seeking long-term stable agricultural production and non-farm employment and by being willing to invest more time, money, and effort in education. There is no significant relationship between education and employment, suggesting that basic education at the primary, junior, and senior high school level for multidimensional relatively poor farmers in central Shaanxi is not able to provide farmers with the experience and competencies needed for traditional agriculture. Furthermore, farmers who are relatively poverty in multiple dimensions face the highest levels of relatively poor in terms of income. These farmers have limited financial resources available for investing in education, primarily due to the heavy financial burden associated with educational expenses. Moreover, they must prioritize healthcare expenditures for their family members before considering higher education. This financial strain, coupled with the inability to support

the determination and resilience of farmers in expanding their production, contributes to the lack of psychological resilience observed among certain groups of farmers.

On the other hand, future value orientation has an amplification effect of 0.397 on farm household life consumption and 0.425 on employment. The decision-making process of farmers is contingent upon their individual experiences and future-oriented choices, ultimately influencing their present and future consumption expenditures. When individuals possess greater confidence in their future prospects, they are inclined to allocate a larger portion of their precautionary savings toward enhancing their quality of life. Consequently, this enables them to transcend the state of relative poverty associated with the current consumption patterns of farmers. The level of consumption of farm households will also have an impact on the channels and efficiency of the household to obtain information, and households with a relatively privileged life can obtain the information they need to be relatively poor through channels such as the Internet, learn new technologies, and thus reduce the relative poverty of information. Therefore, hypothesis H3 has been confirmed.

### 4.3.3. K-Mean Cluster Robustness Test Based on Hierarchical Clustering

To further test the robustness of the above model, to rule out possible data bias and sensitivity to assumption violations, and to improve the reliability of parameter estimates, the farming households are categorized by two indicators: relative poverty in education and relative poverty in income. K-mean clustering was implemented through SPSS statistics 23 to categorize the farmers into high-endowment as well as low-endowment categories. And using the non-parametric hypothesis Mann–Whitney U test, the test *p*-value is significant, indicating that the two types of samples in the level of education and income level of 2 indicators are significantly different. The clustering results are shown in Table 6, the first category of high-education-income endowment farmers has 160 households, accounting for 30.42%, and the second category of low-education-income endowment farmers has 366 households, accounting for 69.58%. The scatter plots of the two clusters are shown in Figure 5.

**Table 6.** Hierarchical clustering results.

| Clustering Category | Frequency | Percentage (%) | Income | Health |
|:---:|:---:|:---:|:---:|:---:|
| High | 160 | 30.42% | 3.40 | 2.32 |
| Low | 366 | 69.58% | 1.52 | 3.42 |
| Total | 526 | 100% | | |

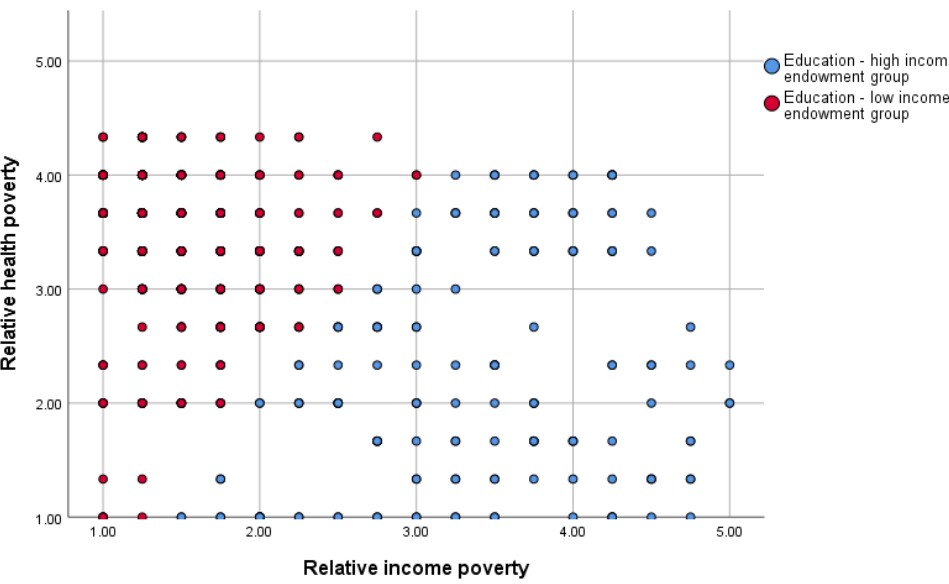

**Figure 5.** Scatterplot of clustering distribution.

A multi-cluster analysis of two types of farmers was conducted to analyze the differences in the multidimensional relative poverty cycle traps as well as the behavior motivation mechanisms of the farmers. The RMSEA for both Model III and Model IV is less than 0.1, and the GFI, AGFI, and CFI are all greater than 0.8, resulting in a better model fit. Table 7 shows that the significance of the path coefficients is basically consistent with Models I and II; the cyclic effect of multidimensional relative poverty for high-endowment farmers is greater than that for low-endowment farmers, and the effect of the behavior motivation mechanism to reduce the cycle of multidimensional relative poverty traps is more obvious. The reason is that health and income are the key mediating dimensions of the multidimensional relative poverty trap, and only when the endowment of health and income is high will the impact among other dimensions be more significant, whereas low-endowment farmers are unable to satisfy their basic needs on the mediating dimensions of health and income, and it is difficult to have a further impact on the other dimensions.

**Table 7.** Results of multi-cluster analysis of high and low endowment farmers.

| Model III | Highly Endowed Group | | Low Endowment Group | |
|---|---|---|---|---|
| | Path Coefficient | *p* | Path Coefficient | *p* |
| IN←NR | −0.018 | 0.843 | 0.421 | *** |
| HE←IN | 0.359 | *** | 0.019 | 0.821 |
| ED←IN | 0.093 | 0.180 | 0.046 | 0.538 |
| HE←CL | 0.040 | 0.799 | 0.068 | 0.277 |
| HE←CL | −0.459 | *** | −0.080 | 0.255 |
| HE←MI | −0.296 | *** | −0.372 | *** |
| ED←EI | 0.646 | *** | 0.056 | 0.383 |
| EM←HE | −0.523 | *** | −0.337 | *** |
| INF←ED | 0.661 | *** | −0.020 | 0.753 |
| EM←ED | 0.325 | 0.001 | 0.142 | 0.012 |
| EM←BE | −0.296 | 0.009 | 0.374 | *** |
| **Model IV** | **Highly Endowed Group** | | **Low Endowment Group** | |
| | Path Coefficient | *p* | Path Coefficient | *p* |
| FO←CL | 0.488 | *** | 0.149 | 0.018 |
| PE←IN | −0.101 | 0.267 | 0.340 | *** |
| IN←ED | 0.226 | 0.007 | −0.062 | 0.461 |
| IN←HE | 0.358 | *** | −0.092 | 0.286 |
| CL←HE | −0.548 | *** | −0.17 | 0.019 |
| PE←ED | 0.428 | *** | 0.208 | 0.002 |
| HE←EM | −0.496 | *** | −0.098 | 0.121 |
| ED←INF | 0.629 | *** | −0.023 | 0.716 |
| FO←EM | 0.076 | 0.345 | 0.571 | *** |

Note: *** is significant at the 1% levels.

## 5. Discussions

The phenomenon of the multidimensional relative poverty trap is influenced by a range of elements, encompassing restricted opportunities for education, healthcare, and jobs, with instances of discrimination and social marginalization. At the farmer level, multidimensional relative poverty traps can occur when farmers lack access to basic services such as health care, education, clean water, and sanitation facilities [44]. At the community level, multidimensional relative poverty traps can occur when entire communities lack access to basic services and opportunities such as education, health care, and employment. The absence of opportunity to obtain necessary resources can result in adverse health outcomes, reduced educational achievements, and restricted economic prospects, hence perpetuating intergenerational poverty. Furthermore, it is important to note that social exclusion and discrimination can serve as additional barriers that restrict the potential for upward socioeconomic mobility and perpetuate the self-reinforcement of poverty within certain communities.

Individuals belonging to impoverished communities may have lower levels of psychological resilience due to a multitude of factors. (1) Insufficient access to fundamental necessities and services: Individuals residing in impoverished conditions may encounter limited access to essential provisions, including sustenance, potable water, adequate housing, healthcare, and educational opportunities. This could potentially restrict their capacity to fulfill their fundamental needs and achieve their aspirations. (2) Constrained economic chances: Individuals residing in impoverished conditions may face restricted access to economic opportunities, including employment prospects and entrepreneurial ventures. This could potentially restrict their capacity to create revenue and enhance their socioeconomic status. (3) Social exclusion and discrimination: Individuals experiencing poverty may encounter social exclusion and prejudice, constraining their ability to obtain resources and avail themselves of opportunities. (4) Mental health issues: Poverty can exert detrimental effects on mental well-being, precipitating conditions such as depression, anxiety, and various other mental health disorders. These factors may potentially intensify sensations of despondency and contribute to a diminished sense of ambition.

Multidimensional groups of relatively poor people can become highly psychologically resilient by developing a sense of collective identity and working together to address the underlying factors that lead to poverty and limited ambition. (1) By building social capital, poorer groups with shorter ambitions can work together, share resources, and knowledge, and support each other to achieve their goals [65]. (2) Encourage collective action and promote the active participation of individuals and groups in development. This includes the formation of community-based organizations, participatory planning processes, and other mechanisms that enable individuals to have a say in how resources are allocated and policies are developed [66]. (3) Promote Leadership Development: Promoting leadership development includes identifying and developing emerging leaders in the community [67]. (4) Mitigating structural barriers: The process of mitigating structural barriers includes the identification and resolution of fundamental problems that contribute to the existence of poverty and hinder the development of aspiration. This may entail the implementation of policies and initiatives aimed at fostering inclusive economic growth, mitigating discriminatory practices, and enhancing accessibility to resources and opportunities [68]. (5) Commemorate accomplishments: Commemorating success entails acknowledging and appreciating the achievements of a community. By engaging in the practice of acknowledging and commemorating accomplishments, socioeconomically disadvantaged communities with limited aspirations can cultivate a collective sentiment of self-esteem and assurance in their capacity to surmount obstacles and attain objectives.

In order to further assist poverty groups with low psychological resilience to escape the multidimensional relative poverty trap, it is important to address the underlying factors that lead to poverty and having limited aspirations with effective strategies, including the following: (1) Facilitate educational and training opportunities: The provision of access to education and training can enable individuals to gain the necessary skills and knowledge required to seek economic opportunities and enhance their overall quality of life. This intervention has the potential to disrupt the perpetuation of poverty and foster a heightened sense of ambition. (2) Fostering entrepreneurship and innovation: The promotion of entrepreneurship and innovation can serve as a catalyst for generating novel economic prospects and stimulating economic advancement. This can potentially offer a means of upward social mobility for people and families with ambitious aspirations, enabling them to escape poverty. (3) Provision of support and resources: The provision of assistance and resources to individuals and families residing in impoverished conditions might contribute to the disruption of the cyclical nature of poverty, hence facilitating social mobility. This may encompass the provision of financial resources, healthcare, and several other social services. (4) It is imperative to engage the communities that are directly impacted by interventions in both the design and implementation processes. This approach ensures that interventions are tailored to address the specific needs and priorities of these groups.

## 6. Conclusions and Implications

Based on a survey of 526 multidimensionally relatively poor households in central Shaanxi, this paper used structural equations to identify the fractal relative poverty traps of farm households and communities and their behavior motivation mechanisms. The study reached the following conclusions: First, there is a self-reinforcing poverty trap in the multidimensional relative poverty of farm households. Second, there is a transmission mechanism between community multidimensional relative poverty and farm household multidimensional relative poverty, which amplifies the overall fractal multidimensional relative poverty trap. Thirdly, the mechanism of behavior motivation of farmers can break the trap of multidimensional relative poverty of farmers. Personal effectiveness stimulates the production will and income-generating drive of farmers, and enhances their confidence in education investment, thus solving the relative poverty of farmers in the dimensions of education, health, income, employment, and information; future value orientation has a stabilizing effect on the production and life of farmers, enabling the proper allocation of resources within limited resources. The future value orientation will have a stabilizing effect on the production and life of farmers, enabling them to allocate resources properly within limited resources and solve consumption and employment poverty.

The study's findings suggest several key insights. Firstly, it is recommended to increase investment in infrastructure development within rural communities located in central Shaanxi. Additionally, it is advised to enhance government expenditure on community education and health care services. These measures aim to effectively address the multidimensional relative poverty trap within the community, taking a top-down approach. Furthermore, individuals can enhance their psychological resilience by forming cohesive groups, cultivating a shared sense of identity, and collaborating to tackle the root causes of poverty and the constraints on their goals. Additionally, it is crucial to enhance educational opportunities and training, foster entrepreneurship and innovation, allocate necessary assistance and resources, and engage impacted communities in the development and execution of strategies aimed at assisting impoverished populations in breaking free from the multidimensional state of relative poverty. Furthermore, there is a pressing need for further research in the crucial field of examining multidimensional relative poverty traps. (1) There is heterogeneity in relative poverty traps at the individual level across different demographics, ages, geographies, and modes of agricultural operation. (2) The operational processes by which community-level multidimensional poverty traps function remain unclear. The involvement of community groups and the creation of policies have a significant impact on the internal dynamics of multidimensional relative poverty within communities, as well as the interactions that occur across different levels. (3) In the future, there will be an expansion of research efforts to encompass a broader area, hence enhancing the generalizability and value of the findings. Ultimately, it is anticipated that this research will serve as a catalyst for generating interest in the examination of rural relative poverty, as well as for the exploration of poverty traps across several levels of analysis.

The operational processes by which community-level multidimensional poverty traps function remain unclear.

**Author Contributions:** Conceptualization, Y.Z.; methodology, Y.Z. and J.H.; software, Y.Z.; validation, Y.Z. and J.H.; formal analysis, Y.Z.; investigation, Y.Z.; resources, Y.Z.; data curation, Y.Z.; writing—original draft preparation, Y.Z.; writing—review and editing, Y.Z. and J.H.; visualization, Y.Z.; supervision, J.H.; project administration, J.H.; funding acquisition, J.H. All authors have read and agreed to the published version of the manuscript.

**Funding:** This research was funded by the National Natural Science Fund of China (Nos. 42075172, 71473196).

**Institutional Review Board Statement:** Not applicable.

**Informed Consent Statement:** Not applicable.

**Data Availability Statement:** The data presented in this study are available within the article.

**Conflicts of Interest:** The authors declare no conflict of interest.

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
