# Peer review of "A Case Study of Farmers’ Behavioral Motivation Mechanisms to Crack the Fractal Multidimensional Relative Poverty Trap in Shaanxi, China"

_agriculture, doi:10.3390/agriculture13112043_

Round 1

Reviewer 1 Report

In my opinion, the article is innovative and has very good content. Its major weakness is not having an research methodology section. There are some ambiguities in the results section, which are mentioned as comments in the attached file. If they are applied, in my opinion, the article is acceptable.

Best Regards,

The English writing quality of the article is good and only needs some minor corrections.

Author Response

Modified Response

Dear Professor.

Those comments are all valuable and very helpful for revising and improving our paper, as well as the important guiding significance to our researches. I am sorry and grateful to you for bringing up several of the issues raised in the essay. Since then, my focus has been solely on revisiting the concerns that you raised. These concerns are discussed in this section, and revisions have been implemented in the manuscript. The main corrections in the paper and the responds to the reviewer’s comments are as flowing:

Question1:This question should be asked as a separate question.

Revise in the article.

Question2: The existence of such a mutual relationship is natural and obvious. Does it really need proof? In my opinion, what is important is the relationship between each dimension, which may be different among societies.

The magnifying effect of poverty is evident in fragile areas, and most studies have looked at the relationship between community and individual poverty mostly in terms of absolute poverty[1,2]. For relative poverty, some cases found a magnifying effect between the two levels[3]. However, when studying relative poverty from a multidimensional perspective, especially when putting the issue of community and individual relative poverty trap interactions into the fractal poverty trap theory, there are fewer studies of this kind and this relationship needs to be proved, and this dynamic relationship is different from case to case[4]. The later part of the paper focuses more on the mechanism of breaking the individual multidimensional relative poverty trap under the influence of community multidimensional relative poverty. I agree with you when you mention that in different societies, or in different circumstances, there may be different relationships. As far as the study area of this paper is concerned, the survey respondents are located in the rural areas of the Loess Plateau which are far away from the cities, and the amplification effect is more obvious with the simple production structure of the farm households and the lower degree of industry. For different environments, different community government strategic choices, as well as natural conditions of different communities, the internal role of fractal traps are different, and there are heterogeneity between the different areas, which I will examine separately in detail in the next study.

Question3:Why are the first letters of the words in the first sentence capitalized?

Revise in the article.

Question4:These should be transferred to the methodology section.

The title here should be Methodology. I apologize and have changed it in the text.

Question5:Mentioning only the abbreviated names of the variables does not help the reader. You should have mentioned the full title of the variables in the table.

Revise in the article.

Question6:Why are your measurement models formative and not reflective?

The model is indeed reflective structural equation modeling and using Amos can only handle reflective structural equation modeling. A mistake was made when graphing with VISIO, and Changes were made to the incorrect modeling diagram.

Explained in the article:

Based on the covariance matrix of variables, structural equation modeling can effectively analyze the structural relationship between latent variables that cannot be directly observed, and it has gradually become one of the important research methods in economics and management. The types of indicators in structural equation modeling usually include reflective and formative types, but if reflective and formative indicators are misused, it will lead to bias in parameter estimation.Reflective structural equations need to satisfy five criteria for modeling: (1) Causality needs to be from latent variables to observed variables. (2) The observed variables must be internally consistent. (3) Observed variables need to be moderately to highly correlated. (4) A latent variable must have at least 3 observed variables. (5) Removing a particular observed variable from a latent variable does not affect the significance of the latent variable.In this paper, changes in the poverty level of some dimensions will lead to changes in the corresponding poverty characteristics of individuals or communities, for example, a bad business environment will affect the development of local rural institutions, community rural cooperatives, and agricultural machinery sharing organizations, which is contrary to the principle of formative structural equations, i.e., changes in latent variables will not lead to changes in the observed variables. Therefore, with reference to the above criteria, the relationship between observed variables and latent variables in this paper is more applicable to formative measurement models.

Question7: In addition, if the coefficients are standard, the values more than one means that there is an error in your model.

The problem about coefficients greater than 1 is that the picture (Figure 3 and Figure 4) in the text are labeled with unstandardized coefficients (unstandardized coefficients are allowed to be greater than 1), which are generally labeled as standardized coefficients in the study, and are corrected here to standardized coefficients.The coefficients between latent variables are also replaced with standardized coefficients in Tables 5. The coefficients in Table 7 are standardized coefficients, so they do not need to be changed . The analysis of amplification effects is also replaced with standardized coefficients in the results.

Question8:You should have explained these methods and their calculation in the methodology section.

K-mean cluster analysis methods and calculation has been added to the essay.

Question9:In my opinion, the existence of this subtitle is not necessary.

Delete the title.

Thanks again for reading this revised response.

Best regards

References:

  1. Brandl, K.; Moore, E.; Meyer, C.; Doh, J. The impact of multinational enterprises on community informal institutions and rural poverty. J Int Bus Stud2022, 53, 1133-1152, doi:10.1057/s41267-020-00400-3.
  2. Boaduo, N. Psychological Implications of Participatory Community Development Projects and their Relevance for Poverty Alleviation in Rural Communities in Africa. J Psychol Afr2010, 20, 209-210, doi:10.1080/14330237.2010.10820366.
  3. Meng, Y.; Lu, Y.Q.; Liang, X.P. Does Internet use alleviate the relative poverty of Chinese rural residents? A case from China. Environ Dev Sustain2023, doi:10.1007/s10668-023-03531-3.
  4. Barrett, C.B.; Swallow, B.M. Fractal poverty traps. World Dev2006, 34, 1-15, doi:https://doi.org/10.1016/j.worlddev.2005.06.008.

Reviewer 2 Report

The topic discussed, multidimensional aspects of poverty, is of particular interest and topicality. It is well known, in fact, that rural poverty is a very important aspect of poverty since it is estimated that three quarters of poverty is concentrated in rural areas of the world.

The methodological approach chosen (Structural Equation Models, SEM) seems to me to be correct, in the light of the need to model complex structures of causal relationships between latent variables from a set of real indicators, i.e. the result of direct investigation. The research questions are well posed.

I propose some additions/adjustments to improve the paper if you wish:

·      -   you have not thought about the relationship between population dispersion with respect to the vastness of the territory (which fuels marginalisation and social distance) to age and type of root or links with agriculture, livestock farming or forestry activity, which are also dispersed throughout the territory, elements that hinder the possibility of organising oneself into social groups capable of expressing one's situation to political decision-makers and the rest of society;

·       -  in addition to underemployment (in the case of self-employment in small businesses), precarious employment and low wages (in the case of employees, especially if seasonal), have you not thought of the other forms of vulnerability affecting young people and children (especially if in large families), women, especially if single and elderly, ethnic groups, minorities and immigrants?

·        -  have development policies been implemented in the survey area?

·         - On line 288 (p. 7) more than results seems to me to be materials and methods

·         - Transport infrastructures, other than those considered in the study exacerbate the problems of poverty.

·         - The limitations of the study and future research should be added at the end of the paper

Good work and congratulations

Author Response

Dear Professor.

Thank you very much for reading my article and giving your valuable suggestions based on where it might be able to expand my research. These comments are highly informative for the next step of our dissertation.

  1. There is heterogeneity in relative poverty traps at the individual level across different demographics, ages, geographies, and agricultural models. Such grouped studies can further draw different research conclusions and create new value.
  2. The articulation of interests by community organizations and policy formulation may influence the internal role of multidimensional relative poverty at the community level, as well as interactions across levels. In particular, the internal role of multidimensional relative poverty at the community level is something that has not been examined in detail in this article.
  3. If the multidimensional relative typology of poverty traps in different regions can be studied in the light of geographic characteristics, infrastructure and transportation, the research hypothesis can be further proved and the conclusions of the study can be more generalized and valuable.

The points not mentioned in the above study, which are also the limitations of the study and the direction of future research, I put in the last part of the article. The article also has some problems, such as the title of the methodology section misspell as Result, and some other details have been corrected in the article. The explanation of reflective structural equations as well as k-mean clustering has been further expanded in the methodology section.

Thank you again for these valuable comments and read it again.

Best regards.